# Development of Two-Tube Loop-Mediated Isothermal Amplification Assay for Differential Diagnosis of *Plasmodium falciparum* and *Plasmodium vivax* and Its Comparison with Loopamp™ Malaria

**DOI:** 10.3390/diagnostics11091689

**Published:** 2021-09-16

**Authors:** Mudsser Azam, Kirti Upmanyu, Ratan Gupta, Karugatharayil Sasi Sruthy, Monika Matlani, Deepali Savargaonkar, Ruchi Singh

**Affiliations:** 1Molecular Biology Laboratory, ICMR- National Institute of Pathology, New Delhi 110029, India; azammudsser@gmail.com (M.A.); kirtiupmanyu@gmail.com (K.U.); sruthychinnu1661995@gmail.com (K.S.S.); 2Department of Paediatrics, VMMC and Safdarjung Hospital, New Delhi 110029, India; ratangupta100@yahoo.com; 3Department of Microbiology, VMMC and Safdarjung Hospital, New Delhi 110029, India; monikamatlani@yahoo.com; 4ICMR-National Institute of Malaria Research, New Delhi 110077, India; dr.deepali27@gmail.com

**Keywords:** LAMP assay, *Plasmodium falciparum*, *Plasmodium vivax*, malaria, differential diagnosis, nested PCR

## Abstract

To strengthen malaria surveillance, field-appropriate diagnostics requiring limited technical resources are of critical significance. Loop-mediated isothermal amplification (LAMP) based malaria diagnostic assays are potential point-of-care tests with high sensitivity and specificity and have been used in low-resource settings. *Plasmodium vivax*–specific consensus repeat sequence (CRS)-based and *Plasmodium falciparum*–specific 18S rRNA primers were designed, and a two-tube LAMP assay was developed. The diagnostic performance of a closed-tube LAMP assay and Loopamp™ Malaria Detection (Pan/Pf, Pv) kit was investigated using nested PCR confirmed mono- and co-infections of *P. vivax* and *P. falciparum* positive (*n* = 149) and negative (*n* = 67) samples. The closed-tube Pv LAMP assay showed positive amplification in 40 min (limit of detection, LOD 0.7 parasites/µL) and Pf LAMP assay in 30 min (LOD 2 parasites/µL). Pv LAMP and Pf LAMP demonstrated a sensitivity and specificity of 100% (95% CI, 95.96–100% and 89.85–100%, respectively). The Loopamp^TM^ Pan/Pf Malaria Detection kit demonstrated a sensitivity and specificity of 100%, whereas Loopamp^TM^ Pv showed a sensitivity of 98.36% (95% CI, 91.28–99.71%) and specificity of 100% (95% CI, 87.54–100%). The developed two-tube LAMP assay is highly sensitive (LOD ≤ 2 parasite/µL), demonstrating comparable results with the commercial Loopamp™ Malaria Detection (Pf/pan) kit, and was superior in detecting the *P. vivax* co-infection that remained undetected by the Loopamp™ Pv kit. The developed indigenous two-tube Pf/Pv malaria detection can reliably be used for mass screening in resource-limited areas endemic for both *P. falciparum* and *P. vivax* malaria.

## 1. Introduction

Malaria is a mosquito-borne acute febrile infection caused by the protozoan parasite *Plasmodium*, which affects 3.4 billion of the world’s population. In 2019, around 229 million malaria cases were reported worldwide, causing a toll of 409,000 deaths [1]. India currently accounts for 4% of the global malaria burden and about 86% of all malaria deaths in the WHO Southeast Asian region [1]. Under the Asia Pacific Leaders Malaria Alliance (APLMA), India has set a target to eliminate malaria by 2030, and efforts for intensified malaria control and elimination are underway in low- and high-transmission areas. Universal access to malaria diagnosis and treatment and strengthened surveillance to accelerate elimination is greatly emphasized [2].

In India, the most prevalent species associated with human morbidity and mortality is *P. vivax* (primarily in plain areas) and *P. falciparum* (in forested and peripheral areas). *P. ovale* and *P. malariae* infections are also reported in some parts of the country with milder symptoms [3,4]. A non-uniform (different) treatment regimen for *P. vivax* and *P. falciparum* makes species-level identification a critical step for patient care and transmission control. 

In the elimination phase, the number of cases decreases substantially, with each case detection becoming essential to prevent perpetual transmission. Though the conventional method for the diagnosis of malaria by light microscopy (sensitivity of 50–500 parasites/μL) remains the gold standard in most clinical settings [5], misdiagnosis due to low parasite count or mixed infections, inadequate quality control and lack of well-trained microscopists remains a significant limitation [6,7,8,9]. The widely used malaria rapid diagnostic tests (RDTs), which rely on the detection of parasite antigen (like HRP2, pLDH, p-aldolase) or host antibody [10], are limited due to false-positive results as an outcome of antigen persistence and false negatives due to gene deletions [11,12]. The nucleic acid amplification methods (conventional PCR, nested PCR and real-time PCR) are of primary importance to accurately identify the malaria parasite but have limited application in field conditions. These expensive and complex methods require trained staff, technically sophisticated and expensive instruments and reagents, and post-PCR analysis. 

An indigenous, highly sensitive field-ready assay, detecting foci of infection in a way timely enough to enable treatment as well as benefit antimalarial drug efficacy monitoring, vaccine studies and screening of vulnerable populations, is perceived as a major priority for malaria elimination. The Loop-mediated isothermal amplification (LAMP) provides an alternative for rapid, sensitive, simple, cost-effective and point-of-care (POC) diagnostic assay with minimal need for trained individuals in malaria-endemic areas. Several LAMP assays have been developed for genus and species-specific identification of malaria infections (Table 1). The developed assays utilize different species-specific gene targets, including the 18S rRNA gene, mitochondrial genes (cytochrome b, cox1 and cytochrome oxidase) and the apical membrane antigen-1 (AMA-1) gene. Fluorimetric (employing SYBR Green or Calcein), chromogenic (hydroxynaphthol blue, phenol red), and naked-eye detection methods present assay sensitivity in the range of 83.3–100% and specificity of 85–100% (Table 1).

The commercially available malaria LAMP kits Loopamp^TM^ Malaria and Illumigene^®^ have been applied in different settings, such as low-transmission areas, and used for screening of imported malaria in non-endemic settings and malaria in pregnancy [29,36,37]. The Loopamp^TM^ Malaria Pan/Pf/Pv Detection kit is an in vitro diagnostic test for the qualitative DNA detection of *Plasmodium* genus (Loopamp^TM^ Malaria Pan), *P. falciparum* (Loopamp^TM^ Malaria Pf) and *P. vivax* (Loopamp^TM^ Malaria Pv). The Loopamp^TM^ Malaria Detection kit has been recognized worldwide and is included in the WHO policy brief on the diagnostics of malaria in low-transmission settings. The applicability of these kits in various settings, the type of samples assessed and comparative diagnostic accuracy have been reviewed recently [38]. The wide range of sensitivity and specificity observed during the evaluations in different regions ([38]; Table 1) and the high cost inferred due to the international import demands an indigenous assay suitable for extensive screening. In the present study, a two-tube based LAMP assay was developed as a POC test for differential diagnosis of *P. falciparum* and *P. vivax* and compared against the commercially available Loopamp^TM^ Malaria Pan/Pf/Pv Detection kit.

## 2. Materials and Methods

### 2.1. Study Area and Sample Collection

The study was conducted according to a detailed protocol that conforms to the STARD (Standards for the Reporting of Diagnostic Accuracy Studies) guidelines. Ethics clearance was obtained from the institutional ethics committees of (a) VMMC and Safdarjung Hospital, New Delhi, India (IEC/VMMC/SJH/Project/January/2018/1029) and (b) ICMR—National Institute of Malaria Research, New Delhi, India (ECR/NIMR/EC/2019/253). Patients with febrile illness reporting at OPD, Safdarjung Hospital, New Delhi, India, and at fever clinic, National Institute of Malaria Research, New Delhi, India, were involved in the study. Venous blood was collected in tubes containing anticoagulant heparin for routine diagnosis of malaria during 2018 and 2019. In this study, a total of 216 blood samples from 149 microscopy confirmed malaria cases (*P. vivax* (*n* = 117); *P. falciparum* (*n* = 28) and cases of mixed infection of *P. falciparum* and *P. vivax* (*n* = 4); total *N* = 149) and 67 microscopy confirmed malaria parasite negative cases (comprising of typhoid (*n* = 34), visceral leishmaniasis (*n* = 25), tuberculosis (*n* = 2) and healthy control (*n* = 6) samples) were analyzed.

### 2.2. Nested PCR

DNA was extracted from 200 µL of blood samples using a QIAamp DNA Blood Mini Kit (QIAGEN, Hilden, Germany) as per the manufacturer’s protocol. For the current study, nested polymerase chain reaction (nested PCR) was considered as the gold standard test and performed on 216 clinical samples as described previously [18,39]. Briefly, a 25 µL reaction was performed for each sample for nest-1 PCR amplification, containing 3 µL of extracted DNA template, 250 nM of each genus-specific primer, 200 µM dNTPs, 2.5 µL of 10X PCR Buffer, 4 mM MgCl_2_, and 1.25 units of Taq Polymerase (New England Biolabs, Ipswich, MA, USA). Five microliters of nest-1 PCR amplification product was used as DNA template for each 25 µL of nest-2 PCR amplification reaction containing species-specific primers. The amplified products of nest-2 PCR were visualized by gel electrophoresis in ethidium bromide stained 2% agarose gel. Reaction with no DNA template having nuclease-free water served as negative control. The samples with positive amplification in nested PCR were considered as true positive, and samples without amplification were considered as true negative [18,39]. 

### 2.3. LAMP Primer Design

The species-specific LAMP primer sets were designed using the Primer Explorer tool (http://primerexplorer.jp/e/), a program for designing LAMP primers. The LAMP primers were designed corresponding to consensus repeat sequence (CRS) Pvr47 of *P. vivax*, present in 14 copies per parasite. The CRS Pvr47 was selected as the target gene as it exhibited higher sensitivity compared to 18S rRNA-based conventional and multiplex PCR assay. The CRS Pvr47 is a sub-telomeric >300 bp repeat with a minimum distance of 100 bp between the multiple copies, bypassing any potential complications in PCR amplification [40,41]. For *P. falciparum*, LAMP primer sets targeting the 18S rRNA gene, previously reported by Han et al., were selected [14].

### 2.4. Construction of Recombinant Plasmid 

Recombinant plasmid with *P. vivax* CRS insert and *P. falciparum* 18S rRNA gene sequence insert was prepared as a control for the establishment of LAMP assay. The *P. vivax* specific CRS and 18S rRNA sequence of *P. falciparum* were amplified, respectively, with 10 ng of template DNA. The amplified products were cloned into the pGEM^®^-T Easy Vector (Promega, Madison, WI, USA) and verified by Sanger sequencing. 

### 2.5. Closed-Tube LAMP Assay 

The closed-tube LAMP assay was performed as described previously [18]. Briefly, the reaction was performed in 25 μL of reaction mixture containing 1 to 2.5 µM each of FIP and BIP primers, 0.15 to 0.25 µM each of F3 and B3 primers, 0.5 to 1 µM each of FLP and BLP primers, 1.4 mM of each dNTP, 0.8 M betaine, 20 mM Tris-HCl (pH 8.8), 10 mM KCl, 10 mM (NH_4_)_2_SO_4_, 8 mM MgSO_4_, 0.1% Triton X-100, 8 units of warm start *Bst* DNA polymerase (New England Biolabs, Ipswich, MA, USA), and 3 μL of DNA sample, independently for *P. vivax* and *P. falciparum*. One μL of 1:10 diluted SYBR™ Green I (Invitrogen, Thermo Fisher Scientific, Eugene, USA) was placed on the inner side of the tube lid, and closed tubes were incubated in a dry bath for the amplification reaction. At the end of the reaction, the tubes were allowed to cool down to room temperature, and a brief spin was given to allow the mixing of SYBR Green I with the amplified product. The reaction mixture with positive amplification instantaneously turned green, while the negative without amplified DNA remained orange. The reaction conditions were standardized for primer concentrations, incubation time (20–60 min) and temperature (60–65 °C).

The results were read by two independent interpreters; discordance, if any in interpretation, was resolved by the assessment of a third interpreter.

### 2.6. Analytical Sensitivity and Specificity 

Tenfold serial dilution (10^8^ to 10 copies of plasmid/µL) of recombinant plasmids containing the target sequence of *P. falciparum* and *P. vivax* was prepared and tested by SYBR™ Green I based Pf LAMP and Pv LAMP assay to determine the analytical sensitivity. For analyzing the specificity of the *P. vivax* specific LAMP assay, other species malaria parasite DNA, other disease control samples (patients of typhoid, tuberculosis, and visceral leishmaniasis) and healthy human control DNA were used. Specificity of 18SrRNA based Pf LAMP have been reported previously [14].

### 2.7. Validation of LAMP Assay with Clinical Samples 

*P. vivax* and *P. falciparum* LAMP assays were validated using 149 microscopy confirmed malaria parasite positive blood samples that were further confirmed by nested PCR for speciation of Plasmodium parasites and 67 confirmed malaria negative cases. Prototypical STARD diagram to report flow of participants through the study for Two-tube Pv/Pf LAMP assay is shown in Appendix A.

### 2.8. Loopamp^TM^ Malaria Pan/Pf/Pv Detection Test

The Loopamp^TM^ Malaria Pan/Pf/Pv Detection kit is a qualitative in vitro diagnostic test that explicitly detects DNA of Plasmodium species causing malaria, extracted from infected human blood samples. The Loopamp^TM^ Malaria Pan Detection kit comprises Malaria Pan (genus)-specific primers, designed specifically to detect the mitochondrial DNA of the four most widespread *Plasmodium* species (*P. falciparum*, *P. vivax*, *P. ovale* and *P. malariae*). The Loopamp^TM^ Malaria Pf Detection kit and Loopamp^TM^ Malaria Pv Detection kit contain Pf-specific and Pv-specific primers that detect the target mitochondrial DNA specific for *P. falciparum* and *P. vivax* sequences, respectively. The cap of the reaction tube is equipped with the strand displacement *Bst* DNA polymerase, dNTPs, Calcein, reaction buffers and Malaria Pan(genus)-specific primers/*P. falciparum* (Pf)-specific primers/*P**. vivax* (Pv)-specific primers in vacuum-dried form. The assay was performed in a batch of 24 reactions comprising 22 test samples, with PC MALARIA (provided in the kit) as positive control and NC MALARIA (nuclease-free water) as a negative control. For the assay, 30 µL of diluted DNA (1:6) extracted from the patient blood sample was transferred into reaction tubes containing dMALPan, dMALPf or dMALPv dried reagents. Sealed reaction tubes were flicked and allowed to stand upside down for 2 min to reconstitute the vacuum-dried reagents. After a short spin to collect the reaction mixture, tubes were incubated at 65 °C for 40 min for DNA amplification. Post amplification tubes were observed in illuminator and in-house UV light detection units. Tubes with light green fluorescence were interpreted as having positive amplification and tubes with no fluorescence as having no amplification. Prototypical STARD diagram to report flow of participants through the study for Loopamp™ Pan/Pf kit and Loopamp™ Pv kit is shown in Appendix A, respectively.

### 2.9. Statistical Analysis

The percentage specificity and sensitivity of Pv LAMP, Pf LAMP and Loopamp^TM^ Malaria (Pan/Pf/Pv) Detection kit were calculated as follows:
Sensitivity = true positives/(true positives + false negatives) × 100.
Specificity = true negatives/(true negatives + false positives) × 100. 

In addition, 95% Confidence Intervals (95% CI), positive predictive values and negative predictive values were calculated using MEDCALC^®^.

## 3. Results

### 3.1. Validation of Clinical Sample by Nested PCR

Among the 149 malaria-infected samples tested, 115 of 117 microscopy confirmed *P. vivax* samples were identified as *P. vivax* mono-infection, 22 of 28 of *P. falciparum* as *P. falciparum* mono-infection; 12 samples were identified as *P. falciparum* and *P. vivax* co-infection against four co-infection confirmed by microscopy. No amplification specific to *P. falciparum* and *P. vivax* sequences was observed in visceral leishmaniasis (*n* = 25), typhoid (*n* = 34), tuberculosis (*n* = 2) and healthy control (*n* = 6) samples.

### 3.2. Primers and Standardization of LAMP Assay

Two sets of LAMP primers specific to consensus repeat sequence of *P. vivax* were successfully designed, and the one that performed better during standardization was used. For *P. falciparum*, previously reported primer targeting 18S rRNA sequence was used [14]. *P. vivax* specific CRS target sequence and *P. falciparum* specific 18S rRNA sequence were successfully cloned in pGEM-T easy vector and used as controls for the optimization of Pv LAMP and Pf LAMP assay. The Pv LAMP assay showed positive amplification at 30 min of reaction incubation at 65 °C with primer concentration of F3/B3 0.2 µM, FIB/BIP 1.3 µM and LF/BF 0.66 µM. For Pf LAMP assay amplification was observed at 40 min of reaction incubation at 65 °C with primer concentration of F3/B3 0.25 µM, FIB/BIP 2 µM and LF/BF 1 µM. Further, for the two-tube based LAMP assay, reactions were performed at 65 °C for 40 min.

### 3.3. Analytical Sensitivity and Specificity of LAMP Assay

The analytical sensitivity of the Pv LAMP assay was determined using 10-fold serially diluted recombinant plasmids (10^8^ to 10 copies/µL) containing the targeted CRS region of *P. vivax*. The assay showed positive results up to 10 copies/µL, equivalent to 0.71 parasites. The Pf LAMP assay showed positive amplification up to 10 plasmids/µL, corresponding to 2 parasites (Figure 1). 

Specificity of the developed Pv LAMP assay was analyzed using DNA isolated from clinical samples infected with *P. vivax*, *P. falciparum*, *Salmonella typhimurium*, *Mycobacterium tuberculosis* and *Leishmania donovani*, along with healthy controls and no template control. The orange to green color change was observed only in the reaction tubes containing *P. vivax* DNA. Other disease samples, healthy controls and no template control remained orange on the addition of SYBR^TM^ Green I, representing a negative Pv LAMP reaction. 

### 3.4. Sensitivity and Specificity of Two-Tube LAMP Assay in Clinical Samples

The Pf LAMP and Pv LAMP assay were evaluated against clinical samples, and the results are summarized in Table 2. Of the 216 clinical samples tested, Pv LAMP assay had positive green fluorescence among 115 *P. vivax* mono-infection samples and 12 mixed (*P. falciparum* and *P. vivax*) infection samples, giving it a sensitivity of 100% (95% CI, 97.14–100%). No fluorescence was observed when Pv LAMP assay was performed with DNA samples of *P. falciparum* (*n* = 22), visceral leishmaniasis (*n* = 25), typhoid (*n* = 34), tuberculosis (*n* = 2) and healthy control (*n* = 6), giving a specificity of 100% (95% CI, 95.94–100%).

For the Pf LAMP reaction, a positive fluorescence was noted in 22 *P. falciparum* and 12 mixed-infection samples, presenting a sensitivity of 100% (95% CI, 89.72–100%). No amplification was observed in 115 *P. vivax* and 67 non-malaria DNA samples, giving a specificity of 100% (95% CI, 97.99–100%).

### 3.5. Loopamp^TM^ Malaria Detection Kit Performance

Loopamp^TM^ Malaria Pan Detection kit was tested against 44 samples (*P. falciparum* (*n* = 15), *P. vivax* (*n* = 15), co-infection (*n* = 8) and other disease (*n* = 6)) (Table 3). A positive fluorescence was obtained in all 38 malaria-infected samples, giving it a sensitivity of 100% (95% CI, 90.78–100%). No amplification was found with non-malaria infected human blood samples, giving it a specificity of 100% (95% CI, 54.07–100%). Loopamp^TM^ Malaria Pf Detection kit reactions were performed against the same batch of 44 samples, and a positive amplification was noted among 15 *P. falciparum* and eight mixed infection samples, giving the kit a sensitivity of 100% (95% CI, 85.18–100%). No amplification was observed in 15 *P. vivax* and six other disease samples, giving the Loopamp^TM^ Malaria Pf Detection kit a specificity of 100% (95% CI, 83.89–100%). Of 88 samples tested using the Loopamp^TM^ Malaria Pv LAMP Detection kit (44 samples of panel-1 used for Loopamp^TM^ Malaria Pan Detection kit and 44 samples of panel-2 comprising 38 fresh *P. vivax* samples and six healthy control samples), 53 samples were detected as true *P. vivax* positive. However, among the eight mixed-infection samples, only seven samples had positive amplification, giving the kit a sensitivity of 98.36% (95% CI, 91.34–99.96%). No fluorescence was observed among the *P. falciparum* (*n* = 15), visceral leishmaniasis (*n* = 3), typhoid (*n* = 3) and healthy control (*n* = 6) DNA samples, giving the Loopamp^TM^ Malaria Pv Detection kit a specificity of 100% (95% CI, 87.23–100%). Cross-tabulation of index tests (Visual LAMP and Loopamp^TM^) results by the results of reference standard test (Nested PCR) for the diagnosis of malaria (*P. falciparum*, *P. vivax* and *Plasmodium* genus) is given in Appendix A.

## 4. Discussion

The Global Technical Strategy for Malaria Elimination 2016–2030 set up by the WHO and the Roll Back Malaria Partnership emphasizes the "high burden to high impact (HBHI)" approach to intensify support for countries with a high burden of malaria. India’s National Vector Borne Disease Control Program (NVBDCP), in collaboration with the WHO, has adopted HBHI approaches in the different states of India for situation analysis. India is progressing towards attaining malaria-free status by 2027 and eliminating the disease by 2030 [1]. To strengthen surveillance and near real-time reporting and monitoring of data to guide better program implementation, the WHO has supported the government in developing the reporting format for malaria under the Integrated Health Information Platform (IHIP). The compromised quality of malaria diagnosis results in significant morbidity and mortality and also contributes to onward transmission, as undiagnosed cases act as an infection reservoir. The LAMP assay for *Plasmodium* parasite detection is a potential point-of-care diagnostic test, as it fulfills all the criteria for a complete commercial sample processing system when special features such as vacuum-dried reaction mixtures sealed within reaction tubes and long-term stability are taken into account. The test is simple, rapid, user friendly and less sophisticated; hence, it can be performed easily in field conditions in low-resource settings. 

This study describes the development of a two-tube lamp assay for the differential diagnosis of *P. falciparum* and *P. vivax* malaria infections. The LOD for Pf LAMP was 2 parasites/µL, and for Pv LAMP, LOD was 0.7 parasites/µL, presenting a detection tool for very low parasitaemia cases. The parasite count of clinical malaria samples tested in this study ranged from 912 to 345,600 for *P. falciparum* and 362 to 79,053 for *P. vivax*. Both the Pf LAMP and Pv LAMP assay had 100% specificity, presenting no cross-reactivity with other diseased patient samples. Our new LAMP assay has a detection time of 40 min. The new consensus repeat based Pv LAMP test developed in this study exhibited sensitivity and specificity of 100% for detection of *P. vivax*, an improvement over other targets both in terms of sensitivity and specificity, and turnaround time of less than one hour. The diagnostic accuracy and sensitivity of the new malaria LAMP assay in this study were higher than those of microscopy and equal to those of nested PCR, with the added advantage of reduced assay time and ease of operation.

The WHO-recognized, commercially available Loopamp^TM^ Malaria Pan/Pf Detection kit has been evaluated by various groups from different countries. A diagnostic accuracy study carried out on febrile returned travelers with this kit reported a sensitivity and specificity of 97.0% and 99.2%, respectively, for the Loopamp^TM^ Malaria Pan test and 98.4% and 98.1%, respectively, for the Loopamp^TM^ Malaria Pf test [36]. At the same time, field assessment of the kit in a remote clinic of Uganda presented a sensitivity of 97.8% for Loopamp^TM^ Malaria Pf test using samples with a *P. falciparum* qPCR titer of ≥2 parasites/μL [37]. However, when tested in low-transmission areas, the sensitivity of the Loopamp^TM^ Malaria Pan and Loopamp^TM^ Malaria Pf was observed as 83.3% and 60%, respectively, with specificity of 99.7% [29]. These results indicate that a highly sensitive target remains to be found for detection of Plasmodium infection in low-transmission settings and in cases with low parasitaemia.

In this study, the Loopamp^TM^ Malaria Pan Detection kit demonstrated a sensitivity and specificity of 100%. Against the same batch of 44 samples, Loopamp^TM^ Malaria Pf Detection kit reactions showed a sensitivity of 100% and specificity of 100%. For the 88 samples tested by the Loopamp^TM^ Malaria Pv Detection kit, a specificity of 100% and a sensitivity of 98.36% was observed, where all the 53 true *P. vivax* samples were detected. However, only seven samples had a positive amplification among the eight co-infection (*P. vivax* and *P. falciparum*) samples. The undermined sample was a mixed infection case confirmed by the nested PCR that went undetected by microscopy, probably due to difficulty in detecting low *P. vivax* parasitemia, as the *P. vivax* often has a lower parasite density (typically 10 times lower) than *P. falciparum* [42]. On the other hand, the two-tube LAMP assay successfully detected all 53 true *P. vivax* samples and eight co-infection (*P. vivax* and *P. falciparum*) samples. The cases of high-density infections show symptoms, and the patients approach health facilities for treatment. However, low-density infections in the community develop very few symptoms; thus, patients seek no treatment, and the infection goes undetected. These low-density infections, below the detection threshold of microscopy and RDTs, continue to infect the vector flies and sustain transmission [43,44]. The developed Pv LAMP assay combined with Pf LAMP assay demonstrated high sensitivity with low LOD, enabling the detection of low parasitemia infections, and may help curb community transmission. 

The different LAMP assays developed using different genus or species-specific gene targets presented sensitivity in the range of 83.3–100% and a specificity of 85–100% (Table 1). When performed with calcein as a detection agent (a low-cost detection agent) or detected based on turbidity with the naked eye, the reactions were compromised with low sensitivity and specificity. 

The SYBR Green I based two-tube test developed here, using 18S rRNA (*P. falciparum*) and CRS (*P. vivax*) as targets, demonstrated high sensitivity (100%) and specificity (100%). It successfully detected *P. vivax* in a co-infection case that remained undetected using the commercially available Loopamp^TM^ Malaria Pv Detection kit. The CRS Pvr47 target deployed in Pv LAMP of our assay is a non-coding, sub-telomeric and highly conserved repeat sequence, without any internal tandem repeats that could potentially interfere with the PCR amplification [40]. After field application trials, the test developed here could be industrialized as a two-tube malaria detection kit (lyophilized) for mass screening in resource-limited and malaria-endemic areas. The vacuum dried reaction mixture will give long term stability and diverse area applicability (even in remote areas). Differentially colored reaction tubes for *P. falciparum* and *P. vivax* will enable easy handling for simultaneous differential diagnosis of *P. falciparum* and *P. vivax* species. Malaria LAMP provides several advantages, including ease of operation and requirement of minimally trained personnel. The diagnostic test requires only a small blood volume (30–60 µL) and is patient-friendly. The Pv and Pf LAMP reactions (for a batch of 24 samples) were completed within 90 min by a single staff member when the DNA template was available beforehand. The need for a simple isothermal heating block, lower start-up costs, and significantly reduced assay time gives LAMP an edge over nested PCR, which requires three to four operational hours. Though the expenses of LAMP reagents are close to those used for nested PCR, a comparison of both techniques in terms of equipment and labor costs reveals that LAMP would be a more affordable option for laboratories in malaria-endemic countries. The frequently observed contamination issue, giving false-positive results, was prevented in the current assay development by following the closed-tube method and using a separate hood for reaction mixture preparation and addition of DNA sample, respectively. 

With an adaptable high-throughput 96-well-plate format, the assay could be deployed for country-wide surveillance, antenatal screening and drug efficacy monitoring [21,45,46]. Moreover, post–malaria elimination status, the test may have commercial utility for its aptness in terms of cost, sensitivity and robustness, under limited funding and programmatic activities for malaria public health concerns.

## 5. Conclusions

In conclusion, the current challenge of malaria elimination, including the complication of developing resistance, requires simple diagnostics for prompt and accurate detection and treatment of malaria parasites. The applicability of this assay remains to be evaluated under field conditions. The integration of the cost-effective and straightforward DNA template preparation method, ideally avoiding pieces of equipment, into the assay pipeline will make it better suited for mass surveillance studies. The two-tube malaria LAMP test developed here for the primary diagnosis of malaria, with differential diagnoses of *P. falciparum* and *P. vivax* supporting targeted treatment, has advantages over other molecular tests in terms of robustness, applicability and sensitivity. The current test format with certain improvisations may potentially replace other nucleic acid– based methods for the diagnosis of malaria in endemic countries. The test will enable parasite identification in asymptomatic and very low parasite density individuals in field settings for mass screening, case management and transmission control. 

## Figures and Tables

**Figure 1 diagnostics-11-01689-f001:**
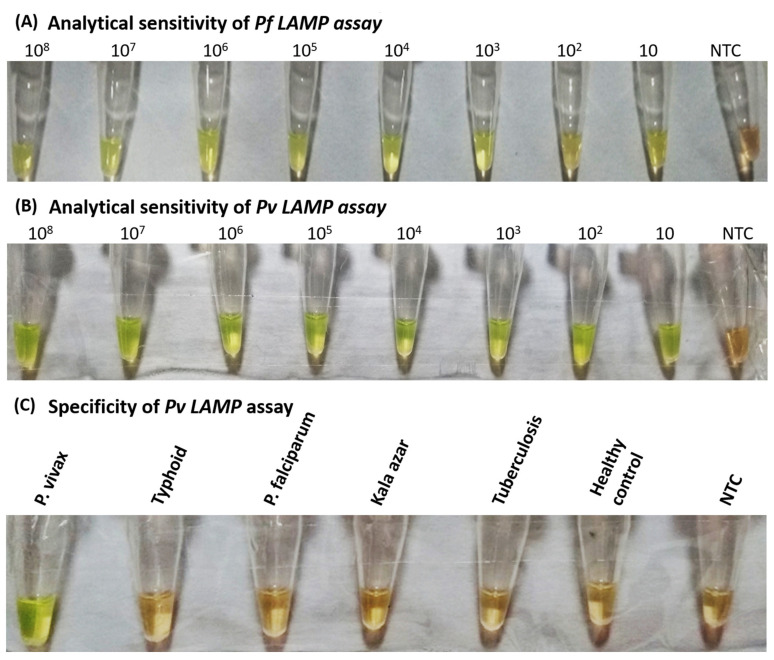
Analytical sensitivity and specificity analysis of SYBR™ Green I based two-tube LAMP assay: (**A**) LAMP assay performed with DNA template as 10-fold serial dilution of plasmid containing 18S rRNA sequence of *P. falciparum*. (**B**) LAMP assay performed with DNA template as 10-fold serial dilution of plasmid containing CRS region of *P. vivax*. (**C**) Pv LAMP assay performed with non-malaria-infected patient samples as DNA template for analyzing the specificity of primers specific for *P. vivax*. NTC-negative test control.

**Table 1 diagnostics-11-01689-t001:** Detailed information of different targets, detection methods and diagnostic accuracy for LAMP assays developed for the diagnosis of malaria.

Target Gene(s)	LAMP Assay	Detection Methods	*Plasmodium* Strain Used	Clinical Sample/Extraction Method	Sensitivity	Specificity	Time and Temperature for Positivity	Reference(s)
18S Ribosomal RNA gene sequence	OneStep turbidimetric conventional LAMP assay	Turbidimetric assay; naked eye	*P. falciparum*, *P. vivax*, *P. malariae*, *P. ovale*, *P. knowlesi. yoelii*	Heat-treated/extracted DNA from clinical blood samples	95–98.5%	94.3–99%	60 °C for 30–120 min	[13,14]
	OneStep SYBR Green conventional LAMP assay	Fluorimetric assay-SYBR Green I and UV light; naked eye	*Plasmodium* spp., *P. falciparum*, *P. vivax, P. malariae*	Extracted parasite DNA from peripheral/placental blood, saliva and/or urine	88.9–100%	90–100%	60–65 °C for 30–100 min	[15,16,17,18,19,20,21]
	Malachite Green-LAMP WarmStart-LAMP	Colorimetric assay-malachite green dye and UV light; phenol red; naked eye	*Plasmodium* spp., *P. falciparum, P. vivax, P. ovale, P. malariae*	Parasite DNA extracted from clinical blood sample by the boil and spin method	95–100%	100%	63 °C for 60 min	[22,23]
	Real-Time fluorescence LAMP/ Multiplex microfluidic LAMP (mμLAMP)	Real-time fluorescence detector- SYTO-9/SYBR green amplification fluorescence peak/hydroxynaphthol blue (HNB)	*P. vivax*, *P. falciparum*	Clinical blood samples	95–97%	91–100%	62–64 °C for 60–90 min	[24,25,26]
Mitochondrial DNA target (*cox1* genes/*cytochrome oxidase subunit 1* gene/others)	OneStep SYBR Green/Calcein conventional LAMP assay/Loopamp^TM^ Malaria Pan/Pf Detection kit (POC)	Fluorimetric assay-SYBR Green I/calcein and UV light; naked eye	*Plasmodium* spp., *P. vivax*, *P. falciparum*	Dried blood spot (DBS)/venous blood sample	83.3–98%	100%	65 °C for 30–60 min	[27,28,29]
	Illumigene Malaria LAMP	Turbidometric assay	*P. falciparum*, *P. vivax, P. ovale*, *P. malaria, P. knowlesi*	Parasite DNA from whole blood	95%	95%	62–65 °C for 60 min	[30]
	High-Throughput, Loop-Mediated Isothermal Amplification (HtLAMP)	Colorimetric assay-hydroxynaphtholblue (HNB); naked eye	*P. vivax*	Parasite DNA from the whole blood sample.	95%	93%	65 °C for 30 min	[31]
	Real-Time fluorescence LAMP (OptiGene)	Fluorometrically assay-SYBR green; naked eye	*P. vivax*, *P. falciparum*, *P. malariae*, and *P. ovale*	Parasite DNA extracted from dried blood spots/dried saliva spots/urine	90–96.7%	85–91.7%	63–65 °C for 30–90 min	[32]
Apical membrane antigen-1 (AMA-1) gene sequence	OneStep conventional LAMP	Fluorimetric assay- SYBR Green I; SYBR^®^ Safe DNA gel stain and UV light;	*P. knowlesi*	Parasite DNA extracted from blood	100%	100%	64 °C for 60 min	[33]
Apicoplast genome	Conventional LAMP	naked eye	*P. falciparum*	Dried blood spot (DBS) sample	92	97	65 °C for 60 min	[34]
Exported protein 1 (*PfExp1*)	Reverse transcription fluorescence -LAMP	amplification fluorescence peak	*P. falciparum*	RNA from freed parasite pellets	90%	Could not be determined *	68 °C for 60 min	[35]

* The specificity was not tested against the real-time PCR confirmed cases, and the data from true positives and negatives are not available for determining the specificity of the PfEXp1 target.

**Table 2 diagnostics-11-01689-t002:** Diagnostic performance parameters of two-tube LAMP assay for differential diagnosis of *P. falciparum* and *P. vivax* using clinical samples.

Kit	Samples	LAMP Assay				
		Cases Tested (Total)	Cases Positive (Total)	Sensitivity/Specificity (95% CI)	Positive Predictive Value	Negative Predictive Value
Pf specific LAMP	Pf/Mix	22/12 (34)	22/12 (34)	100% (89.72–100%)	100%	100%
	Pv/Other Disease/Healthy Control	115/61/6 (182)	0/0/0 (0)	100% (97.99–100%)		
Pv specific LAMP	Pv/Mix	115/12 (127)	115/12 (127)	100% (97.14–100%)	100%	100%
	Pf/Other Disease/Healthy Control	22/61/6 (89)	0/0/0 (0)	100% (95.94–100%)		

Pf—*Plasmodium falciparum*; Pv—*Plasmodium vivax*; CI—confidence interval.

**Table 3 diagnostics-11-01689-t003:** Diagnostic performance parameters of Loopamp^TM^ Malaria (Pan/Pf/Pv) Detection kit.

Kit	Samples	Nested PCR Results		HUMAN LAMP Results				
		Cases Tested (Total)	Cases Positive (Total)	Cases Tested (Total)	Cases Positive (Total)	Sensitivity/ Specificity (95% CI)	Positive Predictive Value (95% CI)	Negative Predictive Value (95% CI)
Loopamp™ Malaria Pan Detection Kit	Pf/Pv/Mix	15/15/8 (38)	15/15/8 (38)	15/15/8 (38)	15/15/8 (38)	100% (90.75–100%)	100%	100%
	Other Disease	6 (6)	0 (0)	6 (6)	0 (0)	100% (54.07–100%)		
Loopamp™ Malaria Pf Detection Kit	Pf/Mix	15/8 (23)	15/8 (23)	15/8 (23)	15/8 (23)	100% (85.18–100%)	100%	100%
	Pv/Other Disease	15/6 (21)	0/0 (0)	15/6 (21)	0/0 (0)	100% (83.89–100%)		
Loopamp™ Malaria Pv Detection Kit	Pv/Mix	53/8 (61)	53/8 (61)	53/8 (61)	53/7 (60)	98.39% (91.34–99.96%)	100%	96.43% (79.44–99.47%)
	Pf/Other Disease/Healthy Control	15/6/6 (27)	0/0/0 (0)	15/6/6 (27)	0/0/0 (0)	100% (87.23–100%)		

Pf—*Plasmodium falciparum*; Pv—*Plasmodium vivax*; CI—confidence interval.

## Data Availability

The authors confirm that all the data reported is available in the manuscript and the Appendix A.

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
