# Peer review of "Development of Two-Tube Loop-Mediated Isothermal Amplification Assay for Differential Diagnosis of Plasmodium falciparum and Plasmodium vivax and Its Comparison with Loopamp™ Malaria"

_diagnostics, 2021, doi:10.3390/diagnostics11091689_

Round 1
Reviewer 1 Report
In the current manuscript, the authors report a method for malaria surveillance, using as Loop-mediated isothermal amplification (LAMP) approach. The authors demonstrate the diagnostic performance of this assay for Plasmodium falciparum and Plasmodium vivax- based on specific consensus repeat sequences. The results reported are remarkable and potential use for mass screening in resource-limited areas. Overall the manuscript sound, well written, and apprears to be useful tool in the field. Therefore, I encourage the authours to include any large field based studies if already performed or include in future studies to demonstrate its wide scale applicability. Furthermore, the authors may consider describing any limitations of the study in the discussion.
Author Response
In the current manuscript, the authors report a method for malaria surveillance, using as Loop-mediated isothermal amplification (LAMP) approach. The authors demonstrate the diagnostic performance of this assay for Plasmodium falciparum and Plasmodium vivax- based on specific consensus repeat sequences. The results reported are remarkable and potential use for mass screening in resource-limited areas. Overall the manuscript sound, well written and appears to be a useful tool in the field. Therefore, I encourage the authors to include any large field-based studies if already performed or include in future studies to demonstrate its wide-scale applicability. Furthermore, the authors may consider describing any limitations of the study in the discussion.
Response: The limitation of the study is now mentioned in the discussion and conclusion. The need to perform a large field-based study to demonstrate its wide applicability is a limitation and now mentioned. Line 386-389.
Reviewer 2 Report
The manuscript described two tube Loop-mediated Isothermal amplification (LAMP) of P. vivax specific consensus repeat sequence and Plasmodium faliciparum specific 18S rRNA and the test has been compared against commercially available Loopamp malaria kit. The conducted study, its outcome and objective is of important as India moving towards control and eventual elimination by 2030 following newly provided WHO milestones. For surveillance and disease emergence investigation LAMP can be an ideal, sensitive and specific tool for disease management. The manuscript has nicely laid out research strategy and initial statistical analysis to justify the outcome. But, there some major flaws regarding need justification and concluding analogy. Also, there are minor flaws regarding inadequate research methodology description. These are mentioned below:
Major concern:
- introduction: The need for conducting this study has not been provided adequately, especially when Loopamp Malaria kit has been recognized worldwide and in WHO policy brief. Some of the discussion related to shortcomings of Loopamp should be mentioned in the Introduction to inform the readers about the need of this study.
- Discussion and conclusion: Some of the assertion, such as replacing microscopy, usage of their two tube LAMP in different diseases level setting etc. is out of context. The study has been conducted on a small clinical sample set without consideration of different disease settings, cost evaluation, and high cost DNA extraction to mention a few. The discussion and conclusion remarks should be compatible with the conducted study, its limitation, its outcome and appropriate compatibility.
Minor concern:
- Materials and Methods: Line 112-116: Details of DNA extraction is required. Nested PCR description is required. Line 119-120: It will be helpful to have a table with LAMP primer information. Line 121-124: More reasoning is required to justify the usage of CRS Pvr47 consensus repeat sequence. Nature of the repeat sequence, its length and abundance, occurrence at different life stages, chance of its mutation. Especially when it is stated there are only 14 copies per parasite, then what that constitute for the whole genome. Line 146-147: How the discordance had been resolved beyond interpreters, whether there had been any gel electrophoresis, sequencing or any other post-amplification process had been used? Line 155: Information about negative control is required. How the positive and negative test has been determined and what is the Gold standard test has been used to compared against. How the false positive and false negative has been determined? How the contamination issue has been resolved?There is no detail about other species detection via LAMP has been provided.
- Results: Figure 1: The figure is confusing. It is missing proper label (A, B and C), number of tubes are not matching, relevancy for other species has not been provided, and no correlation of X-axis and Y-axis labels.
- Discussion: Study limitation as not been discussed. Please provide a detail about that aspect.
Author Response
Major concern:
- Introduction: The need for conducting this study has not been provided adequately, especially when Loopamp Malaria kit has been recognized worldwide and in WHO policy brief. Some of the discussion related to shortcomings of Loopamp should be mentioned in the Introduction to inform the readers about the need of this study.
Response: As suggested, the last paragraph of the introduction has been modified.
Line 93-96. “The wide range of sensitivity and specificity observed during the evaluations in different regions (ref 38; Table 1) and the high cost inferred due to the international import demands an indigenous assay suitable for extensive screening.”
- Discussion and conclusion: Some of the assertion, such as replacing microscopy, usage of their two tube LAMP in different diseases level setting etc. is out of context. The study has been conducted on a small clinical sample set without consideration of different disease settings, cost evaluation, and high-cost DNA extraction to mention a few. The discussion and conclusion remarks should be compatible with the conducted study, its limitation, its outcome and appropriate compatibility.
Response: As per the reviewer suggestions, we have modified the discussion and conclusion remarks to be compatible with the work presented in the study.
Line:386-389. and line 393.
Minor concern:
- Materials and Methods:
Comment: Line 112-116: Details of DNA extraction is required. A nested PCR description is required.
Response: for the DNA extraction method, a line stating “DNA was extracted from 200 µL of blood samples using a QIAamp DNA Blood Mini Kit (QIAGEN, USA) as per the manufacturer's protocol” was given in methods. As QIAamp DNA Blood Mini Kit from Qiagen is a universally used kit, the detailed protocol can be referred from the manual online.
A detailed method has been added for the nested PCR performed in the study. Line 117-129.
Comment: Line 119-120: It will be helpful to have a table with LAMP primer information. Line 121-124: More reasoning is required to justify the usage of the CRS Pvr47 consensus repeat sequence. Nature of the repeat sequence, its length and abundance, occurrence at different life stages, chance of its mutation. Especially when it is stated there are only 14 copies per parasite, then what that constitute for the whole genome.
Response: 18s rRNA gene (present in 5 copies/parasite genome) is widely exploited as a target for species-specific identification. Our study used CRS Pvr47 (present in 14 copies) target with newly designed primers for improvised sensitivity and specificity. The primer sequences of the PvCRS LAMP assay are not revealed due to institutional policy (to protect Intellectual property rights), and the same was mentioned in the cover letter during initial submission.
However, relevant available information about CRS Pvr47 has now been added to the text. The original reference that first demonstrated the utility of this fragment is cited in the manuscript (Demas et al. Applied genomics: data mining reveals species-specific malaria diagnostic targets more sensitive than 18S rRNA. J. Clin. Microbiol. 2011, 49(7):2411-2418.).
Please see the line: Introduction: 136-138 and Discussion: 355-358
Comment: Line 146-147: How the discordance had been resolved beyond interpreters, whether there had been any gel electrophoresis, sequencing or any other post-amplification process had been used?
Response: In general, no discordance was observed, and two interpreters read the results. No post-amplification process was performed in order to avoid reaction contamination issues. As stated in the Methods section “closed-tube” method was followed where reaction tubes are not opened after DNA amplification.
Comment: Line 155: Information about negative control is required. How the positive and negative test has been determined and what is the Gold standard test has been used to compared against. How the false positive and false negative has been determined? How the contamination issue has been resolved? There is no detail about other species detection via LAMP has been provided.
Response: Relevant information is now added in method section 2.2.
Regarding other species detection, nested PCR using widely used genus and species-specific primers (Singh et al. AJTMH,1999) was performed that does not demonstrate any cross-reactivity. Nested PCR was considered the gold standard.
Please see line 127-129.
- Results:
Comment: Figure 1: The figure is confusing. It is missing proper label (A, B and C), number of tubes are not matching, relevancy for other species has not been provided, and no correlation of X-axis and Y-axis labels.
Response: Fig 1 was redesigned to clearly present the information. The label (A, B and C) were added. Fig 1a and Fig1b represent the analytical sensitivity of Pf and Pv lamp determined using plasmid dilutions (8 tubes), including one no template control (total nine tubes). Hence, Fig 1a and Fig1b have 9 tubes.
However, Fig 1c represents the specificity of Pv LAMP with five non-Pv DNA template, i.e. 1)typhoid sample, 2) P. falciparum, 3) Kala-azar, 4) Tuberculosis and 5) healthy control sample, along with one positive control (Pv DNA template) and one negative control (no template control). Hence, a total of 7 tubes is presented in Fig 1C.
- Discussion:
Comment: Study limitation has not been discussed. Please provide a detail about that aspect.
Response: Limitations of this study are now included in the conclusion. Line 386-390.